# Modified Body Mass Index as a Novel Nutritional and Prognostic Marker in Patients with Cardiac Amyloidosis

**Francesca Dongiglio** [1,*], **Giuseppe Palmiero** [1], **Emanuele Monda** [1], **Marta Rubino** [1], **Federica Verrillo** [1], **Martina Caiazza** [1], **Annapaola Cirillo** [1], **Adelaide Fusco** [1], **Erica Vetrano** [1], **Michele Lioncino** [1], **Gaetano Diana** [1], **Francesco Di Fraia** [1], **Giuseppe Cerciello** [2], **Fiore Manganelli** [3], **Olga Vriz** [4] and **Giuseppe Limongelli** [1,5]

1   Inherited and Rare Cardiovascular Diseases, Department of Translational Medical Sciences, University of Campania "Luigi Vanvitelli"—Monaldi Hospital, 80131 Naples, Italy; giuseppe.palmiero@ospedalideicolli.it (G.P.); emanuelemonda@me.com (E.M.); rubinomarta@libero.it (M.R.); fedeverrillo@gmail.com (F.V.); martina.caiazza@yahoo.it (M.C.); cirilloannapaola@gmail.com (A.C.); adelaidefusco@hotmail.it (A.F.); erica.vetrano@gmail.com (E.V.); michelelioncino@icloud.com (M.L.); gaetanodiana1991@gmail.com (G.D.); fyrent718@gmail.com (F.D.F.); limongelligiuseppe@libero.it (G.L.)
2   Haematology Unit, Department of Clinical Medicine and Surgery, University Federico II, 80131 Naples, Italy; cerciello.emato@gmail.com
3   Department of Neurosciences, Reproductive Sciences and Odontostomatology, University Federico II, 80131 Naples, Italy; fiore.manganelli@unina.it
4   Heart Centre, King Faisal Specialist Hospital and Research Centre, Riyadh 11211, Saudi Arabia; olgavriz@yahoo.com
5   Institute of Cardiovascular Sciences, College of London and St. Bartholomew's Hospital, London EC1A 7BE, UK
*   Correspondence: f.dongiglio@gmail.com

**Abstract:** The nutritional assessment is gaining clinical relevance since cardiac cachexia and malnutrition are emerging as novel markers of functional status and prognosis in many cardiovascular disorders, including cardiac amyloidosis (CA). This study aimed to evaluate the prognostic role of different nutritional indices for cardiovascular mortality in patients with CA and subgroups. Fifty CA patients (26 AL and 24 ATTR wild-type) were retrospectively analyzed. All patients underwent a comprehensive clinical and laboratory evaluation. Conventional body mass index (cBMI), modified BMI (mBMI), new BMI (nBMI) and prognostic nutritional index (PNI) were analyzed. Multivariate regression analysis was performed to identify the association between nutritional and other clinical-laboratory parameters with cardiovascular death. Compared to ATTRwt patients, those with AL showed lower mBMI values. No significant difference was observed for the other nutritional indices. During a median follow-up of 11.2 months, a lower mBMI quartile was associated with worse survival, in both groups. In multivariate analysis, mBMI emerged as an independent predictor for cardiovascular death. This study showed that mBMI is a novel index of malnutrition and an independent risk factor for cardiovascular mortality in patients with CA in both AL and ATTRwt form.

**Keywords:** cardiac amyloidosis; nutritional indexes; prognosis

## 1. Introduction

The nutritional assessment is gaining clinical relevance since cardiac cachexia and malnutrition are emerging as novel markers of functional status and prognosis in many cardiovascular disorders [1], including cardiac amyloidosis (CA) [2]. Cardiac cachexia is a complex clinical syndrome caused by cardiac dysfunction characterized by body composition abnormalities (muscle wasting and peripheral edema) with progressive weight loss, mainly driven by protein-calorie malnutrition. It has a complex multifactorial physiopathology, including inflammation, associated with many chronic disorders and carries a poor quality of life and poor outcome [3,4]. Instead, malnutrition is defined by the imbalance

---



between the cellular supply of nutrients and energy and their demand. Its pathophysiology could be the consequence of increased energy demands, inadequate food intake, a defect in the metabolism of nutritional elements, malabsorption, or a combination [1]. Malabsorption due to gut edema, anorexia due to cytokine production, and limited food intake due to anorexia and fatigue are all mechanisms proposed in malnutrition and subsequent overt cachexia in cardiovascular disorders [5]. Malnutrition and cachexia could coexist. However, they are still different since malnutrition can be easily reversed with adequate nutrition [6,7].

Amyloidosis includes a group of diseases characterized by the accumulation of amyloid fibrils derived from the aggregation of misfolded proteins in the extracellular spaces of different organs, whose function is consequently progressively compromised. The most common proteins which form amyloid fibrils with common cardiac involvement are immunoglobulin light chains (AL) and transthyretin (TTR) [8]. AL amyloidosis results from an uncontrolled proliferation of a single clone of plasma cells (plasma cells dyscrasia), resulting in an overproduction of immunoglobulin AL that deposits as amyloid in many tissues [9,10]. Thus, AL is a systemic disease characterized by a rapidly progressive clinical course with a prognosis largely dependent on toxic-infiltrative cardiomyopathy, consisting of median survival of fewer than six months if left untreated [11]. In ATTR amyloidosis, the misfolded TTR is the protein primarily produced in the liver. The transformation of this 127 amino acid protein into amyloid is stimulated by unknown mechanisms related to aging in wild-type ATTR (ATTRwt) or by at least 120 known point mutations in its gene, resulting in single amino acid substitutions in the hereditary ATTR (ATTRh) [12].

Some retrospective studies identified nutritional abnormalities in about half of AL patients and are associated with impaired functional status and increased morbidity and mortality [13,14]. A recent retrospective analysis of retrospective or prospective, observational or interventional studies focusing on the impact of AL amyloidosis on nutritional status showed that malnutrition was present in about 65% of patients with a negative impact on quality of life and survival [15].

In systemic amyloidosis, many tissues and organs could be involved. Indirect cardiac-driven and direct gastrointestinal (GI) tract involvement was sometimes present and could determine many clinical manifestations, such as anorexia, dysphagia, vomiting, diarrhea, malnutrition and weight loss [16]. However, due to under-recognition and delayed investigation, supportive treatment and preventive strategies (i.e., nutritional support and physical exercise) necessary for maintaining a good nutritional status are not commonly implied in the care of amyloidosis patients [17,18].

Nutritional assessment relies on tools, laboratory tests and questionnaires that can also be performed in non-referral centers. However, the definition of nutritional status is still challenging due to the lack of internationally accepted thresholds for anthropometric and biochemical variables, especially in specific populations, such as in CA.

Several nutritional status indexes have been suggested [19–25]. The modified body mass index (mBMI), calculated as the product of BMI and serum albumin, is an accurate method of measuring the nutritional status of patients with volume overload and those with ATTR (22). In particular, the mBMI overcomes the limitation emerging from its component alone: BMI, which reflects obesity and general physical status but does not reflect fluid balance; serum albumin, which reflects fluid balance but does not provide information on the general physical status. Furthermore, the Prognostic Nutritional Index (PNI) [24], a nutritional status index taking into account serum albumin and total lymphocyte count, is independently associated with long-term survival in patients hospitalized for acute heart failure with reduced or preserved ejection fraction [25].

Therefore, a specific score, the Patient-Generated Subjective Global Assessment (PG-SGA), was used for examining the nutritional status in a cohort of newly diagnosed, treatment naïve patients with AL amyloidosis. A PS-SGA score $\geq 4$ identified malnourished patients requiring specialist nutritional intervention and was associated with poor quality

of life and reduced survival [26]. However, this score has been validated for AL patients and does not apply to ATTR patients.

This study aimed to evaluate the prognostic role of different nutritional indices for cardiovascular mortality in patients with CA and subgroups.

## 2. Materials and Methods

### 2.1. Study Population and Design

Between March 2018 and December 2019, we conducted a retrospective study on 50 patients with CA referred to the Inherited and Rare Cardiovascular Diseases Unit of the Monaldi Hospital (Naples, Italy). The population consisted of 26 patients with AL and 24 with ATTRwt. All AL patients had biopsy-proven CA and positive immune histology for AL amyloidosis. On the other hand, all ATTRwt patients had a non-invasive diagnosis with positive 99mTc-hydroxy methylene-diphosphonate scintigraphy, negative hematological screening and negative genetic testing for TTR gene mutations [27].

All patients have been evaluated after an adequate period of optimal medical therapy to avoid excessive blood pressure and volume fluctuations. Patients with AL CA were stratified in four stages by the revised Mayo Clinic staging system according to the difference between involved and uninvolved light chain (dFLC), N-terminal probrain natriuretic peptide (NT-proBNP) and high sensitivity cardiac troponin I (HS-cTnI) [28,29]. Instead, patients with ATTRwt were stratified in three stages according to NT-proBNP and estimated glomerular filtration rate (eGFR) [30].

### 2.2. Clinical and Laboratory Assessment

All patients underwent an extensive clinical evaluation. Medications and clinical variables, such as New York Heart Association (NYHA) functional class, were recorded. Laboratory parameters [white blood cells (WBC) count, red blood cells (RBC) count, hemoglobin (Hb), creatinine, eGFR, NT-proBNP, HS-cTnI, serum albumin in AL and ATTRwt patients plus dFLC in AL CA patients only] were measured from peripheral venous blood samples using standard commercially available assays. Reference values were as follows: WBC $4.4$–$11.3 \times 103/\mu$L, RBC $4.0$–$5.1 \times 106/\mu$L, Hb $12.2$–$15.3$ g/dL, creatinine $0.55$–$1.10$ mg/dL, eGFR $> 60$ mL/min/1.73 m$^2$, NT-proBNP $< 125$ ng/L, HS-cTnI $< 36.5$–$116.0$ in female and $40.8$–$115.0$ pg/mL in males, serum albumin $3.2$–$4.6$ g/dL.

### 2.3. Neurologic Assessment

Neurological involvement was assessed by investigating the history, signs and symptoms of neuropathy (e.g., carpal tunnel syndrome, spinal stenosis, paresthesia) and dysautonomia (e.g., orthostatic hypotension, urinary bladder disorders, gastrointestinal disorders). Diabetic patients were excluded to avoid bias resulting from diabetic peripheral neuropathy.

### 2.4. Nutrition Assessment

A trained nutritionist performed a nutrition assessment: height and body weight were measured using a balance beam platform scale and an altimeter. Therefore, classical body mass index (cBMI) was calculated by the formula: weight (kg)/height (m$^2$). Furthermore, the new BMI (nBMI), proposed by English mathematicians Nick Thefethen and colleagues, was calculated by the formula: $1.3 \times$ weight (kg)/height (m$^{2.5}$). The modified BMI (mBMI) was calculated by multiplying the classical BMI by the serum albumin expressed in grams per liter [31]. The prognostic nutritional index (PNI) was calculated by the formula: $10 \times$ serum albumin (g/dL) $+ 0.005 \times$ total lymphocyte (count per mm$^3$) [32].

### 2.5. Echocardiographic Assessment

All patients underwent standard transthoracic echocardiography using a Vivid E9 ultrasound system (GE Healthcare, Horten, Norway) equipped with an M5S 3.5-MHz transducer. Two-dimensional, Color-Doppler, Pulsed-wave and Continuous-wave Doppler data were acquired and then stored on a dedicated workstation for offline analysis (EchoPAC

Version 202, GE Vingmed Ultrasound, Horten, Norway). Appropriate echocardiographic windows were obtained in all the patients. Chamber quantification was carried out according to current recommendations [33]. Diastolic parameters were collected in apical four-chamber view by wave and Tissue Doppler [34]. Myocardial contraction fraction (MCF), a volumetric measure of myocardial shortening independent of chamber size and geometry, was calculated as the ratio of stroke volume to myocardial volume [35]. Stroke volume was obtained by the Simpson biplane method, while myocardial volume was calculated as the ratio of LV mass (LVM) over 1.05 (myocardial density). Peak systolic longitudinal strain (LS) and 2D Speckle-Tracking (2D-ST) strain measurements were performed in the three standard apical views, and an automatically generated bullseye of segmental peak systolic LS values was obtained. LV global longitudinal strain (LVGLS) was calculated by the average of regional LS values, as recommended [36]. Regional LS for six basal, six mid and five apical views of the LV were averaged to obtain three mean regional LS values (basal, mid and apical). Relative regional strain ratio (RRSR) was calculated by dividing the average apical LS by the sum of the average basal and mid-LS values [37]. Systolic pulmonary artery pressures (sPAP) were estimated from tricuspid regurgitation jet velocities and inferior vena cava diameters, and respiratory variations. Tricuspid annular plane systolic excursion (TAPSE) was measured with M-mode modality from the tricuspid annular longitudinal excursions in an apical four-chamber view. Right ventricular systolic excursion velocity was assessed, placing the pulsed Tissue Doppler Imaging sample volume in the basal segment of the right ventricle free wall and measuring the highest velocity of the s' wave [38].

*2.6. Statistical Analysis*

All calculations were performed using commercially available statistical software (SPPS Statistics Version 24.0, IBM Corp., Chicago, IL, USA). All continuous variables were presented as mean $\pm$ SD, and a *p*-value of <0.05 was considered statistically significant. Continuous variables were tested for normal distribution using the Kolmogorov–Smirnov test and compared by Student's *t*-test. Discrete variables were presented as percentages. Categorical variables were compared using the $\chi^2$ test. Linear regression analysis was performed to assess variables associated with mBMI.

Moreover, univariate linear regression analysis was used to find the best predictors of survival in CA patients by entering nutritional variables (cBMI, nBMI, mBMI) and a set of variables that are known to be associated with cardiovascular mortality in AL and ATTR amyloidosis (age, NYHA functional class, eGFR, NT-proBNP, HS-cTnI, MCG, LVEF, LVGLS, E/Em ratio, LAVI, TAPSE, s' wave, sPAP). Multivariate analysis was performed by entering into the model a set of variables that were considered significant on univariate analysis for *p*-values < 0.25 to identify the parameters that best predicted the survival in CA. Independent associations were expressed as B co-efficient and 95% confidence intervals. Kaplan–Meier curves were constructed to explore differences in medium-term survival in two different subgroups stratified according to the presence or absence of malnutrition and compared using the long-rank test. The endpoint was cardiovascular death. Survival was calculated from the day of CA diagnosis until the date of death or last follow-up.

## 3. Results

*3.1. Baseline Characteristics*

The clinical features of the examined cohort are presented in Table 1. For the 50 patients with CA, the mean age at diagnosis was 69.2 $\pm$ 14.0 years, and 68% were male. Most of them had HF as deducible from their prevalence in NYHA functional class II (50%) and III (36%), the extensive use of diuretics (94%) for symptoms management and the high serum concentrations of NT-proBNP (median 3153 pg/mL, IQR 1306-9140); 44% of the pooled population had a history of sensory-motor peripheral neuropathy, and 36% had a history of cardiac dysautonomia. Moreover, patients with CA showed a significant increase in cardiac

biomarkers (HS-cTnI: median 145 pg/mL; IQR 63-308) with a moderate decrease in kidney function assessed with eGFR (69.7 $\pm$ 33.1 mL/min/1.73 m$^2$).

**Table 1.** Clinical characteristics of the study cohort. Data are presented in mean $\pm$SD or *n* (%), unless otherwise indicated.

| | CA (*n* = 50) | AL CA (*n* = 26) | ATTRwt CA (*n* = 24) | *p*-Value |
|---|---|---|---|---|
| Baseline clinical and functional parameters | | | | |
| Age, years | 69.2 $\pm$ 14.0 | 60.6 $\pm$ 13.4 | 78.6 $\pm$ 7.1 | 0.0001 |
| Male sex | 34 (68%) | 13 (50%) | 21 (87.5%) | 0.005 |
| NYHA functional class | | | | |
| Class I | 5 (10%) | 3 (11.5%) | 2 (8.5%) | |
| Class II | 25 (50%) | 10 (38.5%) | 15 (62.5%) | 0.260 |
| Class III | 18 (36%) | 11 (42%) | 7 (29%) | |
| Class IV | 2 (4%) | 2 (8%) | 0 (0%) | |
| Peripheral neuropathy | 22 (44%) | 6 (23%) | 16 (66%) | 0.002 |
| Cardiac dysautonomia | 18 (36%) | 7 (27%) | 11 (46%) | 0.136 |
| Cardiovascular death | 15 (30%) | 9 (30%) | 6 (25%) | 0.334 |
| Height, cm | 166.4 $\pm$ 7.9 | 166.9 $\pm$ 9.1 | 165.8 $\pm$ 6.6 | 0.646 |
| Weight, kg | 70.2 $\pm$ 12.1 | 67.9 $\pm$ 12.0 | 72.7 $\pm$ 11.9 | 0.156 |
| BSA, m$^2$ | 1.7 $\pm$ 0.2 | 1.7 $\pm$ 0.2 | 1.8 $\pm$ 0.2 | 0.363 |
| cBMI, kg/m$^2$ | 25.3 $\pm$ 3.9 | 24.3 $\pm$ 3.7 | 26.4 $\pm$ 3.9 | 0.061 |
| nBMI, kg/m$^2$ | 25.5 $\pm$ 4.0 | 24.5 $\pm$ 3.9 | 26.6 $\pm$ 3.9 | 0.062 |
| mBMI | 954.6 $\pm$ 227.9 | 874.7 $\pm$ 216.2 | 1041.1 $\pm$ 211.9 | 0.008 |
| PNI | 70.5 $\pm$ 12.6 | 69.9 $\pm$ 18.8 | 71.2 $\pm$ 8.1 | 0.739 |
| SBP, mmHg | 115.8 $\pm$ 22.4 | 111.9 $\pm$ 24.5 | 120.1 $\pm$ 19.6 | 0.198 |
| DBP, mmHg | 67.8 $\pm$ 12.1 | 66.3 $\pm$ 12.8 | 69.4 $\pm$ 11.4 | 0.377 |
| Stage | | Revised Mayo Clinic | Gillmore | |
| I | | 3 (11.5%) | 11 (46%) | |
| II | | 5 (19%) | 7 (29%) | |
| III | | 8 (31%) | 6 (25%) | |
| IV | | 10 (38.5%) | | |
| Laboratory parameters | | | | |
| RBC, millions/mm$^3$ | 4.5 $\pm$ 0.6 | 4.4 $\pm$ 0.7 | 4.6 $\pm$ 0.6 | 0.342 |
| Haemoglobin, g/dL | 13.2 $\pm$ 1.9 | 12.8 $\pm$ 1.7 | 13.6 $\pm$ 2.1 | 0.199 |
| WBC, x10$^3$/$\mu$L | 6635.0 $\pm$ 2727.9 | 6862.1 $\pm$ 3387.4 | 6389.0 $\pm$ 1805.7 | 0.546 |
| Creatinine, mg/dL | 1.2 $\pm$ 1.1 | 1.2 $\pm$ 1.6 | 1.1 $\pm$ 0.3 | 0.687 |
| eGFR, mL/min/1.73 m$^2$ | 69.7 $\pm$ 33.1 | 80.9 $\pm$ 38.6 | 57.7 $\pm$ 20.7 | 0.012 |
| Serum albumin, g/dL | 3.7 $\pm$ 0.6 | 3.6 $\pm$ 0.6 | 3.9 $\pm$ 0.6 | 0.036 |
| NT-proBNP, pg/mL, median (IQR) | 3153 (1306–9140) | 3617 (1039–) | 3153 (1565–4411) | 0.190 |
| HS cTnI, pg/mL, median (IQR) | 145 (63–308) | 142 (39–319) | 145 (78–303) | 0.301 |
| Medications | | | | |
| Beta-blockers | 5 (10%) | 7 (27%) | 10 (41%) | 0.212 |
| Amiodaron | 17 (34%) | 1 (4%) | 4 (15%) | 0.150 |
| ACE-inhibitors | 10 (20%) | 2 (7%) | 8 (33%) | 0.070 |
| Sartans | 1 (2%) | 0 (0%) | 1 (4%) | 0.480 |
| Diuretics | 47 (94%) | 24 (92%) | 23 (96%) | 0.531 |
| Anti-mineralcorticoids | 18 (36%) | 10 (38%) | 8 (33%) | 0.468 |
| Calcium-antagonists | 1 (2%) | 1 (4%) | 0 (0%) | 0.520 |
| Antiplatelet agents | 8 (16%) | 4 (15%) | 1 (16%) | 0.601 |
| Vitamin K antagonists | 4 (8%) | 3 (11%) | 1 (4%) | 0.336 |
| Direct oral anticoagulants | 17 (34%) | 4 (15%) | 13 (54%) | 0.004 |

Abbreviations: ACE, angiotensin converting enzyme; BSA, body surface area; CA, cardiac amyloidosis; cBMI, conventional body mass index; DBP, diastolic blood pressure; eGFR, estimated glomerular filtration rate; HS-cTnI, high sensitivity cardiac troponin I; mBMI, modified body mass index; nBMI, new body mass index; NT-proBNP, N-terminal probrain natriuretic peptide; NYHA, New York Heart Association; PNI, Prognostic Nutritional Index; SBP, systolic blood pressure; RBC, red blood cells; WBC, white blood cells.

Patients with AL amyloidosis were younger than the ATTRwt form (Age 60.6 ± 13.4 vs. 78.6 ± 7.1 years; $p < 0.0001$). On the contrary, ATTRwt patients were prevalently male (87.5 vs. 50%; $p < 0.005$) and showed a higher rate of peripheral neuropathy (66 vs. 23%; $p < 0.002$). Patients with AL amyloidosis showed lower serum albumin values than ATTRwt patients (3.6 ± 0.6 vs. 3.9 ± 0.6 g/dL; $p < 0.036$). On the other hand, ATTRwt patients showed a lower eGFR (57.7 ± 20.7 vs. 80.9 ± 38.6 mL/min/1.73 m$^2$; $p < 0.012$).

### 3.2. Echocardiographic Assessment

Echocardiographic parameters of the examined cohort are shown in Table 2. Patients with CA showed severe LV concentric hypertrophy with a relatively small cavity size. LV and right ventricular (RV) systolic function was reduced and assessed with classical and deformation parameters. Moreover, LV filling pressure was significantly high with concomitant left atrial (LA) enlargement.

**Table 2.** Echocardiographic parameters of the study cohort.

| | CA (*n* = 50) | AL CA (*n* = 26) | ATTRwt CA (*n* = 24) | *p*-Value |
|---|---|---|---|---|
| | Echocardiographic parameters | | | |
| LVEDD, mm | 42.7 ± 5.9 | 40.9 ± 5.7 | 44.5 ± 5.6 | 0.030 |
| LVESD, mm | 28.6 ± 6.5 | 26.5 ± 6.7 | 30.9 ± 5.6 | 0.016 |
| IVSD, mm | 16.9 ± 3.5 | 16.5 ± 2.8 | 17.4 ± 4.1 | 0.360 |
| PWD, mm | 15.4 ± 3.4 | 14.7 ± 2.9 | 16.1 ± 3.7 | 0.172 |
| RWT ratio | 0.74 ± 0.22 | 0.74 ± 0.19 | 0.74 ± 0.25 | 0.926 |
| LVM, g | 297.6 ± 102.0 | 265.2 ± 85.4 | 332.7 ± 108.6 | 0.018 |
| LVM indexed, g/m$^2$ | 166.7 ± 54.3 | 149.9 ± 42.5 | 184.9 ± 60.5 | 0.021 |
| LVEDV, mL | 73.7 ± 26.9 | 66.7 ± 28.3 | 81.4 ± 23.6 | 0.054 |
| LVESV, mL | 37.2 ± 18.5 | 33.4 ± 21.4 | 41.3 ± 14.2 | 0.136 |
| LVEF, % | 50.7 ± 8.7 | 51.6 ± 8.4 | 49.7 ± 8.9 | 0.449 |
| MCF, % | 13.8 ± 5.3 | 14.1 ± 5.7 | 13.6 ± 4.9 | 0.760 |
| LVGLS, % | −10.5 ± 3.3 | −10.5 ± 3.5 | −10.5 ± 3.1 | 0.961 |
| RRSR | 0.98 ± 0.41 | 0.99 ± 0.36 | 0.97 ± 0.47 | 0.840 |
| E wave, cm/s | 88.6 ± 23.6 | 88.9 ± 28.0 | 88.2 ± 18.2 | 0.916 |
| A wave, cm/s | 55.5 ± 26.9 | 58.6 ± 30.6 | 50.3 ± 19.3 | 0.406 |
| E/A ratio | 3.0 ± 5.9 | 3.6 ± 7.5 | 2.1 ± 0.9 | 0.505 |
| DecT, ms | 151.7 ± 48.3 | 146.1 ± 47.5 | 157.7 ± 49.4 | 0.407 |
| E/e' ratio | 18.1 ± 8.4 | 19.7 ± 10.5 | 16.4 ± 5.0 | 0.175 |
| LAD, mm | 46.8 ± 8.7 | 44.6 ± 7.7 | 49.3 ± 9.3 | 0.055 |
| LAV, mL | 87.1 ± 32.3 | 73.0 ± 23.2 | 102.5 ± 34.1 | 0.001 |
| LAVI, mL/m$^2$ | 48.5 ± 16.3 | 41.4 ± 13.1 | 56.2 ± 16.2 | 0.001 |
| TAPSE, mm | 14.9 ± 5.0 | 15.0 ± 5.4 | 14.9 ± 4.7 | 0.948 |
| St wave, cm/s | 10.9 ± 3.1 | 11.2 ± 3.2 | 10.7 ± 3.0 | 0.577 |
| sPAP, mmHg | 40.3 ± 9.9 | 38.1 ± 8.6 | 42.5 ± 10.9 | 0.118 |

Abbreviations: CA, cardiac amyloidosis; DecT, deceleration time; E/A, E wave/A wave; E/e', E wave/e' wave; IVSD, interventricular septum in diastole; LAD, left atrial diameter; LAV, left atrial volume; LAVI, left atrial volume index; LVEDD, left ventricular end diastolic diameter; LVEDV, left ventricular end diastolic volume; LVESD, left ventricular end systolic volume; LVESV, left ventricular end systolic volume; LVEF, left ventricular ejection fraction; LVGLS, left ventricular global longitudinal strain; LVM, left ventricular mass; MCF, myocardial contraction fraction; PWD, posterior wall in diastole; RRSR, relative regional strain ratio; RWT, relative wall thickness; sPAP, systolic pulmonary artery pressure; TAPSE, tricuspid annular plane systolic excursion.

ATTRwt showed larger LV cavity size (LVEDD 44.5 ± 5.6 vs. 40.9 ± 5.7 mm; $p < 0.030$) and mass (LVM indexed 184.9 ± 60.5 vs. 149.9 ± 42.5 g/m$^2$; $p < 0.021$) and higher LA volume (LAVI 56.2 ± 16.2 vs. 41.4 ± 13.3 mL/m$^2$; $p < 0.001$) than AL patients, without significant differences in LV and RV systolic function, LV diastolic function, and pulmonary artery pressures.

### 3.3. Nutritional Assessment

There were no differences between groups in nutritional indices, except for mBMI, which was lower in AL patients ($24.5 \pm 3.9$ vs. $26.6 \pm 3.9$; $p < 0.008$) due to lower serum albumin levels in this group.

In our cohort, mBMI was significantly associated with age (R 0.963; $p < 0.0001$), sex (R 0.913; $p < 0.0001$), peripheral neuropathy (R 0.680; $p < 0.0001$), cardiac dysautonomia (R 0.573; $p < 0.0001$), cardiac biomarkers (NT-proBNP: R 0.502; $p < 0.028$–HS-cTnI: R 0.484; $p < 0.036$), eGFR (R 0.919; $p < 0.0001$), LV mass (R 0.944; $p < 0.0001$), LV systolic function (LVEF: R 0.955; $p < 0.0001$–MCF: R 0.926; $p < 0.0001$–GLS: R 0.974; $p < 0.0001$–RRSR: R 0.920, $p < 0.0001$), LV diastolic function (LAVI: R 0.942; $p < 0.0001$–E/Em ratio: R 0.907; $p < 0.0001$), RV systolic function (TAPSE: R 0.948; $p < 0.0001$–St: R 0.958; $p < 0.0001$) and pulmonary pressure (sPAP: R 0.955; $p < 0.0001$).

### 3.4. Survival Analysis

After a median follow-up of 11.2 months (IQR 3.7–15.0), 15 of the 50 patients (30%) died, nine patients (30%) in the AL group and six patients (24%) in the ATTRwt group.

Table 3 shows univariate and multivariate regression analysis results of proposed nutritional indices (cBMI, nBMI, mBMI, PNI) with established clinical (age NYHA class), laboratory (NT-proBNP, HS-cTnI, eGFR) and echocardiographic (MCF, LVEF, LVGLS, E/Em ration, LAVI, TAPSE, S' wave, sPAP) prognostic markers of CA proposed in previous reports.

**Table 3.** Univariate and multivariate regression analysis of the proposed nutritional indices with established clinical, laboratory, and echocardiographic parameters.

| | Univariate | | Multivariate | |
|---|---|---|---|---|
| | B (95% CI) | *p*-Value | B (95% CI) | *p*-Value |
| Age, years | 0.998 (0.961–1.036) | 0.908 | | |
| NYHA class | 2.550 (1.221–5.324) | 0.013 | | |
| cBMI, kg/m$^2$ | 0.875 (0.748–1.024) | 0.097 | | |
| nBMI, kg/m$^2$ | 0.862 (0.737–1.007) | 0.061 | | |
| mBMI | 0.996 (0.994–0.999) | 0.001 | 0.996 (0.994–0.999) | 0.006 |
| PNI | 0.995 (0.954–1.039) | 0.835 | | |
| eGFR, mL/min/1.73 m$^2$ | 0.996 (0.979–1.014) | 0.678 | | |
| NT-proBNP, pg/mL | 1.000 (1.000–1.000) | 0.679 | | |
| HS-cTnI, pg/mL | 1.001 (1.000–1.002) | 0.002 | | |
| MCF, % | 0.972 (0.876–1.080) | 0.602 | | |
| LVEF, % | 0.995 (0.935–1.058) | 0.865 | | |
| LVGLS, % | 0.863 (0.715–1.042) | 0.126 | | |
| E/e' ratio | 1.064 (1.014–1.116) | 0.011 | | |
| LAVI, mL/m$^2$ | 1.007 (0.974–1.040) | 0.701 | | |
| TAPSE, mm | 0.912 (0.807–1.031) | 0.142 | | |
| St, cm/s | 0.874 (0.731–1.045) | 0.139 | | |
| sPAP, mmHg | 1.024 (0.978–1.073) | 0.308 | | |

Abbreviations: cBMI, conventional body mass index; E/e', E wave/e' wave; eGFR, estimated glomerular filtration rate; HS-cTnI, high sensitivity cardiac troponin I; LAVI, left atrial volume index; LVEF, left ventricular ejection fraction; LVGLS, left ventricular global longitudinal strain; mBMI, modified body mass index; MCF, myocardial contraction fraction; nBMI, new body mass index; NT-proBNP, N-terminal probrain natriuretic peptide; NYHA, New York Heart Association; PNI, Prognostic Nutritional Index; sPAP, systolic pulmonary artery pressure; TAPSE, tricuspid annular plane systolic excursion.

NYHA functional class (β 2.550; *p* < 0.013), mBMI (β 0.996; *p* < 0.001), HS-cTnI (β 1.001; *p* < 0.002) and E/Em ratio (β 1.064; *p* < 0.011) were significantly associated with cardiovascular mortality in the study population. Multivariate analysis was introduced into the model, along with a set of variables that were considered significant on univariate analysis for *p*-values < 0.25 (NYHA functional class, cBMI, nBMI, mBMI, HS-cTnI, LVGLS, E/Em ratio, TAPSE, s' wave). In the multivariate analysis, mBMI emerged as an independent predictor for adverse cardiovascular events (β 0.996, IC95% 0.994–0.999; *p* < 0.006). In both AL and ATTRwt group, the lowest mBMI quartile was associated with the worst cumulative survival (Figures 1–3).

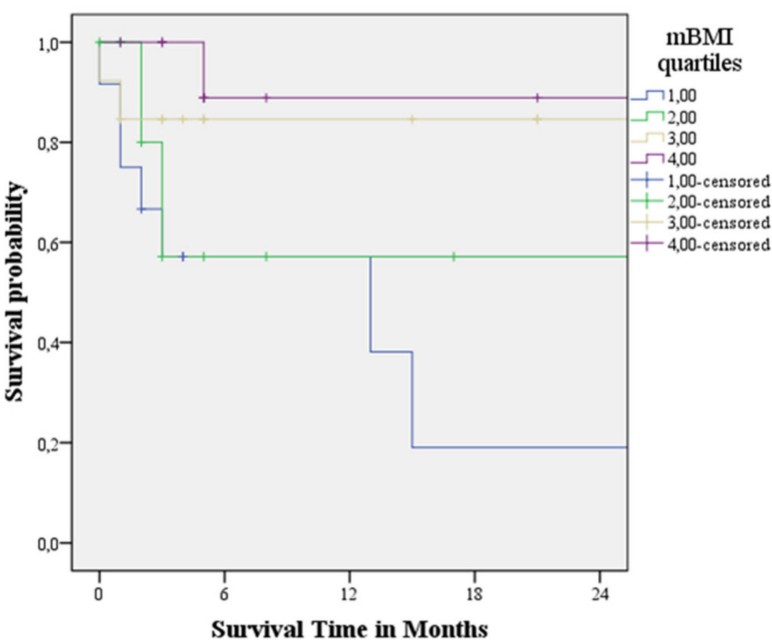

**Figure 1.** Kaplan-Meier survival curve analysis of cardiovascular mortality in the overall population across the quartiles of the modified Body Mass Index.

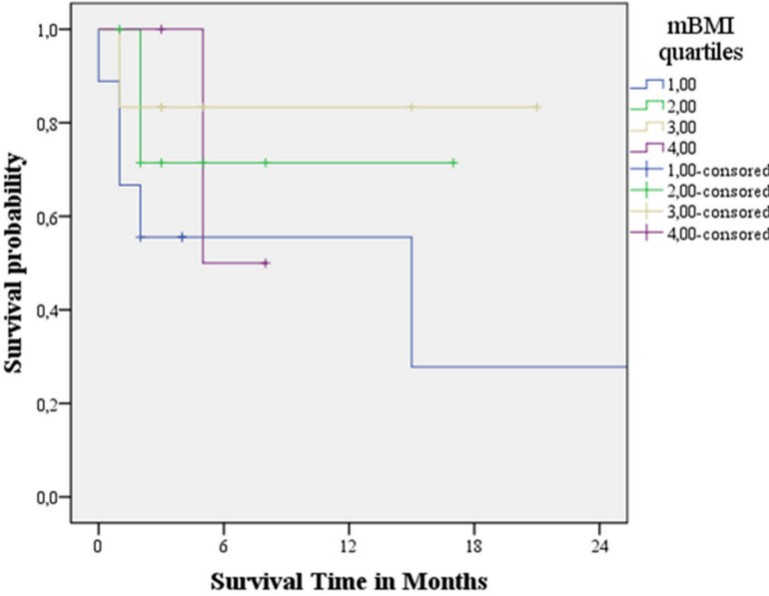

**Figure 2.** Kaplan-Meier survival curve analysis of cardiovascular mortality in AL group across the quartiles of the modified Body Mass Index.

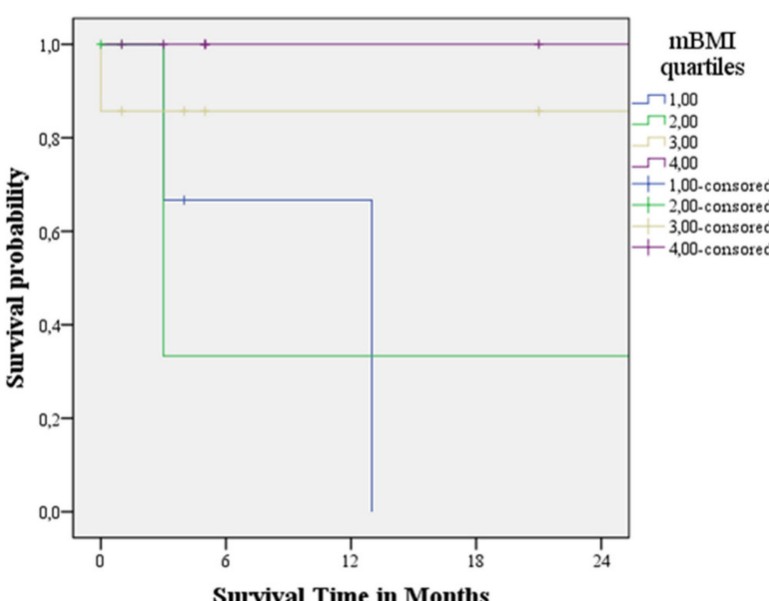

**Figure 3.** Kaplan-Meier survival curve analysis of cardiovascular mortality in ATTR group across the quartiles of the modified Body Mass Index.

## 4. Discussion

This is the first study comparing mBMI with other nutritional indices in predicting cardiac adverse events in a cohort study of patients with both AL and ATTRwt amyloidosis. Indeed, malnutrition measured by mBMI has already been correlated with patient survival in both ATTRh and ATTRwt [19,39], and lower BMI values have been associated with a worse prognosis in renal AA amyloidosis [40]. However, no studies are reported the role of mBMI in AL amyloidosis.

In the present study, the nutrition status, assessed by mBMI in patients with CA, is associated with medium-term mortality in AL and ATTRwt forms. Indeed, mBMI revealed more prognostic predictive power than clinical, laboratory and echocardiographic variables commonly associated with survival in CA. Moreover, mBMI reduction reflects the more advanced stages of the disease due to its significant association with parameters of LV (HS-cTnI, NT-proBNP, LVM, RRSR, LVEF, MCF, GLS) and RV involvement (TAPSE, s' wave, sPAP). This result is consistent with previous data in literature suggesting that in CA, as in HF, malnutrition is a frequent feature in patients affected, especially in those with more advanced stages of the disease and could be associated with cardiac cachexia [41].

The malnutrition pathophysiology in CA may involve many mechanisms. Chronic overload might be an indirect cause of malabsorption, abnormal intestinal microcirculation, reduced gut perfusion, and endotoxin-induced increased mucosal permeability, which results in local edema. This mechanism already described in HF could be translated to CA patients [41,42]. Indeed, in this study, the mBMI was associated with other nutritional correlates, such as hemoglobin and total lymphocyte count, and with variables associated with impaired cardiac performance, such as LV filling (E/Em ratio) and pulmonary (sPAP and RAA, surrogate of central venous pressure pressures) and RV systolic function (TAPSE and s' wave), confirming the relationship between chronic overload, RV dysfunction and intestinal congestion [43,44].

Furthermore, in AL, as in ATTRwt forms, there could be a direct GI involvement by the infiltrative disorder. Some retrospective cohort studies have reported that GI amyloidosis (defined by the presence of GI symptoms along with direct biopsy verification) is present in less than 10% of patients. On the contrary, in some prevalence studies, the GI involvement in ATTRh amyloidosis ranged between 56% and 69% of patients. However, in the latter, the prevalence was assessed only on the presence or absence of GI symptoms [45]. Nonetheless, additional extensive prospective studies are needed to assess the real burden of GI involve-

ment in CA due to the paucity of data. GI involvement in systemic amyloidosis seems to be a sequential process. The amyloid deposits start in the vascular wall, followed by the mucosa and the muscular layers and are finally involved in the nervous fibers of the enteric plexus, causing dysmotility [46]. It seems prevalent in the small intestine; however, any organ could be potentially involved. Common clinical manifestations included symptoms of impaired gastric emptying (nausea, vomiting, postprandial fullness) and symptoms of bowel dysmotility (diarrhea, constipation, abdominal pain and fecal incontinence) [47].

Apart from the loss of tissue function directly affected by amyloid infiltration, cardiovascular autonomic dysfunction may contribute to malnutrition. It causes intestinal mucosal dysfunction by redistributing the blood flow away from the splanchnic circulation [41], altered motility by the myenteric plexus's direct involvement, and reduced secretory function [48]. In this study, the mBMI was significantly associated with cardiovascular autonomic dysfunction, underlying the possible role of mBMI in reflecting malnutrition due to multiple and complex mechanisms.

Moreover, we hypothesize that the differences in mBMI between AL and ATTRwt CA may be partly attributable to inflammation due to the direct cytotoxic effect exerted by the circulating immunoglobulin free-light chains on the GI tract. However, dedicated studies are needed to confirm this hypothesis.

## 5. Limitations

This study is a single-center experience conducted on a relatively limited number of patients. There were significant differences in age and sex between AL and ATTR groups, potentially influencing the patient's metabolism and nutritional parameters. Moreover, we assessed malnutrition due to CA without any information on cardiac cachexia due to a lack of specific inflammatory parameters, such as uric acid. Finally, most patients enrolled in the study were in advanced stages of the disease, as suggested by the reduced LVEF values and the increased cardiac biomarkers levels, representing a possible bias in translating our results to patients at the early stages of the disease.

## 6. Conclusions

This study demonstrated that mBMI is a novel index of malnutrition and an independent risk factor for cardiovascular mortality in patients with CA in both AL and ATTRwt form. Since it is readily available, mBMI could be a useful screening tool for malnutrition in all forms of CA. Malnutrition is a reversible syndrome that could be efficaciously treated with an adequate nutritional supply. For this reason, mBMI is a promising and valuable index in recognizing early CA patients who may benefit from specific nutritional treatments.

**Author Contributions:** F.D., G.P., E.M., M.R., F.V., M.C., A.C., A.F., E.V., M.L., G.D., F.D.F., G.C., F.M., O.V. and G.L. contributed to conceptualization, methodology, software, validation, formal analysis, investigation, resources, writing—original draft preparation, writing—review and editing, visualization. All authors have read and agreed to the published version of the manuscript.

**Funding:** This research received no external funding.

**Institutional Review Board Statement:** The study was conducted in accordance with the Declaration of Helsinki, and approved by the Ethics Committee of University of Study of Campania "Luigi Vanvitelli" (protocol code AOC/0034419/2021—26 November 2021).

**Informed Consent Statement:** Informed consent was obtained from all subjects involved in the study. Written informed consent has been obtained from the patients to publish this paper.

**Conflicts of Interest:** The authors declare no conflict of interest.

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
