# Peer review of "Modified Body Mass Index as a Novel Nutritional and Prognostic Marker in Patients with Cardiac Amyloidosis"

_cardiogenetics, doi:10.3390/cardiogenetics12020017_

Round 1

Reviewer 1 Report

Dear authors,

Major comments:

Interesting and well-documented subject.

1. You mention that "However, the impact of the different nutritional status indexes on outcomes have not yet been systematically evaluated in a well-characterized cohort of CA patients." and “However, no studies are reporting the role of mBMI in AL amyloidosis.”

Actually, there follow research studies (2) and a review focusing on the impact of nutritional indices in cardiac amyloidosis patients and analyzing data from 110 to 300 patients. Such articles should be (re)considered for reasoning, rephrasing, and/or commenting on compatible findings or discrepancies.

Prayman T. Sattianayagam et al. “A prospective study of nutritional status in immunoglobulin light chain amyloidosis” ©2013 Ferrata Storti Foundation. This is an open-access paper. doi:10.3324/haematol.2012.070359

Elissa Driggin et al. Markers of nutritional status and inflammation in transthyretin cardiac amyloidosis: association with outcomes and the clinical phenotype. AMYLOID

https://doi.org/10.1080/13506129.2019.1698417

Shana Souza Grigoletti et al. “Focused review on nutritional status of patients with immunoglobulin light chain amyloidosis” https://doi.org/10.1016/j.currproblcancer.2021.100833

Minor comments:

1. Line 87: Please rephrase: “...optimal medial therapy...”

2. Some parts require english language improvement,

for example in Introduction and in lines 240, 291, 294.

Author Response

Dear reviewer,

We greatly appreciate your comments and tried to follow your suggestion. About this:

  1. We have rephrased the aim of the study. In this study, we want to compare the impact on the outcome of different nutritional indexes applicable in both forms of cardiac amyloidosis (AL and ATTR). For this reason, we have not considered other indexes or scores that can be applied only in one subgroup (f.e. the PG-SGA score in AL amyloidosis)
  2. We have cited the PG-SGA score (Sattianayagam PT, Lane T, Fox Z, et al. A prospective study of nutritional status in immunoglobulin light chain amyloidosis. Haematologica. 2013;98(1):136-40) to underline its role in nutrition evaluation and prognosis in AL amyloidosis.
  3. The work by Drigging et al. was already cited in the first draft for its importance in introducing mBMI in ATTR patients. However, as pointed out by you, the message was not clear and has been clarified in the revisions.
  4. As suggested, we have tried to improve the English language throughout the text.

Reviewer 2 Report

It is convenient that when defining an abbreviation in the writing of the article at first from then on, they use the abbreviation only, for example for AL CA, it is the same case for ATTRwt CA.

It is convenient that when defining an abbreviation in the writing of the article at first from then on, they use the abbreviation only, for example for AL CA, it is the same case for ATTRwt CA.

The present paper is relevant and interesting in research about Modified Body Mass Index as a Novel Nutritional and Prognostic Marker in Patients with Cardiac Amyloidosis, this topic is original and this paper give more details about the prognostic role of different nutritional indices for cardiovascular mortality in patients with CA and subgroups, because patients with cardiac amyloidosis have an increased risk of sudden cardiac death.

The paper is well written, and the text is clear and easy to read.

Author Response

Dear reviewer,

we sincerely thank you for your comments. We followed your recommendation about the use of adequate abbreviation using AL and ATTR appropriately (mention their different use in different settings: f.e. in differentiating the two subgroups of the study or differentiating the two entities in the background and in discussion